# Impact of Starch Binding Domain Fusion on Activities and Starch Product Structure of 4-α-Glucanotransferase

**DOI:** 10.3390/molecules28031320

**Published:** 2023-01-30

**Authors:** Yu Wang, Yazhen Wu, Stefan Jarl Christensen, Štefan Janeček, Yuxiang Bai, Marie Sofie Møller, Birte Svensson

**Affiliations:** 1Enzyme and Protein Chemistry, Department of Biotechnology and Biomedicine, Technical University of Denmark, DK-2800 Kongens Lyngby, Denmark; 2School of Food Science and Technology, Jiangnan University, Wuxi 214122, China; 3Protein Chemistry and Enzyme Technology, Department of Biotechnology and Biomedicine, Technical University of Denmark, DK-2800 Kongens Lyngby, Denmark; 4Laboratory of Protein Evolution, Institute of Molecular Biology, Slovak Academy of Sciences, SK-84551 Bratislava, Slovakia; 5Department of Biology, Faculty of Natural Sciences, University of SS. Cyril and Methodius, SK-91701 Trnava, Slovakia; 6Applied Molecular Enzyme Chemistry, Department of Biotechnology and Biomedicine, Technical University of Denmark, DK-2800 Kongens Lyngby, Denmark

**Keywords:** 4-α-glucanotransferase, starch binding domain (SBD) fusion, starch modification, tandem SBDs, glycoside hydrolase family 77 (GH77), carbohydrate binding module family 20 (CBM20)

## Abstract

A broad range of enzymes are used to modify starch for various applications. Here, a thermophilic 4-α-glucanotransferase from *Thermoproteus uzoniensis* (TuαGT) is engineered by N-terminal fusion of the starch binding domains (SBDs) of carbohydrate binding module family 20 (CBM20) to enhance its affinity for granular starch. The SBDs are N-terminal tandem domains (SBD_St1_ and SBD_St2_) from *Solanum tuberosum* disproportionating enzyme 2 (*St*DPE2) and the C-terminal domain (SBD_GA_) of glucoamylase from *Aspergillus niger* (*An*GA). In silico analysis of CBM20s revealed that SBD_GA_ and copies one and two of GH77 DPE2s belong to well separated clusters in the evolutionary tree; the second copies being more closely related to non-CAZyme CBM20s. The activity of SBD-TuαGT fusions increased 1.2–2.4-fold on amylose and decreased 3–9 fold on maltotriose compared with TuαGT. The fusions showed similar disproportionation activity on gelatinised normal maize starch (NMS). Notably, hydrolytic activity was 1.3–1.7-fold elevated for the fusions leading to a reduced molecule weight and higher α-1,6/α-1,4-linkage ratio of the modified starch. Notably, SBD_GA_-TuαGT and-SBD_St2_-TuαGT showed *K*_d_ of 0.7 and 1.5 mg/mL for waxy maize starch (WMS) granules, whereas TuαGT and SBD_St1_-TuαGT had 3–5-fold lower affinity. SBD_St2_ contributed more than SBD_St1_ to activity, substrate binding, and the stability of TuαGT fusions.

## 1. Introduction

4-α-glucanotransferases (4αGT, EC 2.4.1.25), belonging to the glycoside hydrolase family 77 (GH77) (http://www.CAZy.org, accessed on 23 December 2022) [1], catalyze four different reactions: cyclization, coupling, hydrolysis, and disproportionation [2]. The disproportionation is attractive as it involves a transfer of malto-oligosaccharides to suitable α-1,4-glucan acceptors. When the α-1,4-glucan acceptor is the α-glucan chain of the covalent enzyme-intermediate, a circular molecule is formed, named a large-ring cyclodextrin (LR-CD), by connecting the reducing and non-reducing ends [3]. When the acceptor in the disproportionation reaction is a different α-1,4-glucan chain, the transfer of a fragment to its non-reducing end can lead to elongation of exterior chains in branched α-glucan molecules [4].

Starch binding domains (SBDs), as a special group of carbohydrate binding modules (CBMs), provide numerous starch-active enzymes with enhanced affinity for different α-glucans [5]. Among the 94 CBM families (http://www.cazy.org/, accessed on 23 December 2022) [1], 15 were defined as SBDs, namely CBM20, 21, 25, 26, 34, 41, 45, 48, 53, 58, 68, 69, 74, 82, and 83 [5]. SBDs can have important affinity for α-glucans—including granular starches [6,7], show micromolar affinity for β-cyclodextrin (a starch model) [8,9], and are thought to be able to disentangle α-glucan chains of double helixes on the starch granule surface [5,8,9,10] offering an explanation for their stimulation of granular starch hydrolysis. Still, the main function of SBDs is considered to be molecular recognition and binding to starch granules. SBDs thus facilitate the reaction of the catalytic domains (CDs) by bringing the active site in close contact with substrate [11]. SBDs can also guide the α-glucan chain to be modified to the active site crevice on the CD [12].

The aim of the present work is to confer a thermophilic starch-modifying 4-α-glucanotransferase from *Thermoproteus uzoniensis* (TuαGT) [13] with novel functional properties by one-by-one fusion with three different SBDs, two from *Solanum tuberosum* (potato) disproportionating enzyme 2 (*St*DPE2) of the glycoside hydrolase family 77 (GH77) [14] and one from *Aspergillus niger* glucoamylase (*An*GA) of GH15 [15]. The effect on the different types of GH77 activities as obtained in the three fusions SBD_St1_-TuαGT, SBD_St2_-TuαGT, and SBD_GA_-TuαGT was analysed by using maltotriose, amylose, gelatinised normal and waxy maize starches, and native waxy maize starch granules as substrates. In general, SBD-fusion increased the activity of TuαGT on amylose and gelatinised starch, but reduced the disproportionating activity on maltotriose. The SBD-TuαGTs had an increased affinity for granular starch but only slightly changed the chain length distribution of gelatinised NMS. The three SBDs exerted individual effects on the function of TuαGT. Especially, SBD_St1_ and SBD_St2_ showed different influences on the thermostability and binding affinity of TuαGT, suggesting that tandem SBDs from *St*DPE2 individually play different functional roles. Lastly, SBD-fusion can be a promising technology to change the substrate specificity and activity of enzymes.

## 2. Results and Discussion

### 2.1. 4-α-Glucanotransferase SBD Fusions

Several 4-α-glucanotransferases have been reported to contain starch binding domains (SBDs) [5,13]. To improve starch affinity and modification for TuαGT, three different fusion proteins were constructed by attaching SBDs of the family CBM20 to the N-terminus of the enzyme. Two SBDs from *Solanum tuberosum* disproportionating enzyme 2 (*St*DPE2) [14] (SBD_St1_, the N-terminal, and SBD_St2_, the second in tandem), and one (SBD_GA_) from *Aspergillus niger* glucoamylase (*An*GA) [15] were used (Figure 1). The fusions of the CD and SBDs were performed via an 18-residues linker (TTGESRFVVLSDGLMREM) that naturally connects the SBD_St1–_SBD_St2_ tandem with the CD in *St*DPE2 (Figure 1).

### 2.2. Bioinformatics Analysis

In order to put the three above-mentioned experimentally fused SBD_St1_, SBD_St2_, and SBD_GA_ into the overall context of the CBM20 family, 65 different starch hydrolases and related enzymes were selected for in silico analysis (Table 1). The emphasis was mainly on GH77 DPE2s, both from *Eukaryota* (including the *St*DPE2) and *Bacteria*, known to contain two recognizable CBM20s [16]. The set to be analysed was completed by various well-known CBM20s from amylolytic enzymes classified into several CAZy families (including *An*GA) as well as several non-CAZymes, such as phosphoglucan, water dikinase (GWD3), laforin, genethonin-1, etc. [5,16,17,18,19].

From the 65 selected enzymes, it was possible to sample 87 CBM20 sequences (see Table 1 for details). It is worth mentioning that although there was a stretch in almost each DPE2 sequence (regardless the bacterial or eukaryotic origin) for two CBM20 copies at the N-terminus, only those not lacking most of the known CBM20 functionally important binding site residues [8,9,12] were taken into the analysis. Interestingly—based on a detailed inspection of their amino acid sequences—the hypothetical DPE2s from *Linum tenue* (GenBank Acc. No.: CAI0439830.1) and *Ricinus communis* (UniProt Acc. No.: B9SCF0) obviously contain only one CBM20 copy (data not shown). It is of note that of the two potential starch binding sites of CBM20, only starch binding site one, being formed by Trp543, Lys578, and Trp590 (GH15 *A. niger* glucoamylase numbering [8]), is well conserved (Appendix A), whereas residues forming starch binding site two may vary [5], as evidenced by the structural complexes of CBM20s from GH15 *A. niger* glucoamylase with cyclodextrin (Tyr527, Tyr556 and Trp563) [8] and GH13_2 *Bacillus circulans* cyclodextrin glucanotransferase with maltose (Tyr633 and Trp636) [19]—having only the tryptophan (Trp563 vs Trp 636) conserved (Appendix A). Of the SBD_St1_, SBD_St2_, and SBD_GA_ used in the present study, only SBD_GA_ from GH15 *A. niger* glucoamylase, that possesses all the key residues involved in binding (Appendix A), was previously demonstrated to bind starch [8]. SBD_St1_ and SBD_St2_ each lack one of the conserved residues at starch binding site one—the SBD_St1_ lysine (Lys578; *A. niger* GH15 CBM20 numbering) and the SBD_St2_ tryptophan (Trp590)—and only the tryptophan (Trp563) of starch binding site two is conserved in both; however, SBD_St1_ might have a stronger ability to bind since it has a tryptophan corresponding to Tyr527 at binding site two (Appendix A).

The evolutionary tree (Figure 2), constructed from the sequence alignment, illustrated several facts: (i) each of the two CBM20 copies from GH77 DPE2s forms its own cluster; (ii) all CBM20s from other CAZymes cluster together (including SBD_GA_ of *An*GA; cluster B) and separately from both groups covering the two CBM20 copies of GH77; (iii) the second CBM20 copy of GH77 DPE2s (including SBD_St2_ of *St*DPE2; cluster D) exhibits a closer relatedness to CBM20s from non-CAZymes (such as GWD3, laforin, genethonin-1, etc.; cluster C) than to those from other CAZyme families (cluster B); and (iv) the clade of the first CBM20 copy of GH77 DPE2s (including SBD_St1_ of *St*DPE2, cyan in Figure 2) covers also the second and the third CBM20 copies from laforins from *Cyanidioschyzon merolae* and *Chondrus crispus,* respectively, [18] (brown clade in cluster A, Figure 2) as well as the CBM20 from the four-domain GH13_2 cyclodextrin glucanotransferase from *Nostoc* sp. PC9229 [20] (green in cluster A, Figure 2). The results from the bioinformatics analysis thus indicate that the three CBM20s studied here, i.e., SBD_St1_, SBD_St2_, and SBD_GA_, are positioned in three different clusters of the evolutionary tree (Figure 2) and may confer the parental enzyme TuαGT distinctly different biochemical properties by the fusion.

### 2.3. Biochemical Properties of TuαGT and SBD-TuαGT Fusions

The produced TuαGT, SBD_St1_-TuαGT, SBD_St2_-TuαGT, and SBD_GA_-TuαGT migrated in SDS-PAGE as single protein bands estimated to 56, 68, 67, and 69 kDa (Figure 3A), respectively, in agreement with the theoretical values (see Section 3.5). The optimal reaction temperature and pH for the maltotriose disproportionation activity were around 70 °C and 7.0 for the different forms of TuαGT (Figure 3B,D). However, SBD_GA_-TuαGT had a lower temperature optimum of 60 °C (Figure 3B). This is in good agreement with previously reported pH and temperature optima for the total activity on amylose and maltose of TuαGT at 6.0 and 75 °C [13]. TuαGT was nearly 100% active at 80 °C, indicating it is a thermophilic enzyme, which also showed significantly reduced activity at <60 °C. Notably, all three SBD-TuαGT fusions were relatively less active than TuαGT at >70 °C, but more active at <60 °C (Figure 3B). The improved affinity to starch of the SBD-fusions (see Section 2.4) may contribute to their relatively higher activity than the parent enzyme TuαGT at <60 °C, whereas the lower relative activity of the fusions at >70 °C may stem from their poorer thermostability as illustrated by the time progress for the loss of activity at 50 °C (Figure 3C). Notably, after 20 h at 50 °C, the parent TuαGT maintained ~35% activity. However, all SBD-TuαGT fusions lost more activity than TuαGT during the first 5 h at 50 °C and SBD_St1_-TuαGT and SBD_GA_-TuαGT retained only about 20% activity after 8 h, whereas SBD_St2_-TuαGT kept remarkably ~65% of its activity after 20 h (Figure 3C). Improved thermostability was previously found by N-terminal fusion of a CBM1 to β-mannanase from *Aspergillus usamii* YL-01-78 (reAuMan5A-CBM), having a temperature optimum at 75 °C compared with 70 °C for wild-type (reAuMan5A), indicating a stabilizing effect of the CBM1 on the CD [21]. In another study, Wang et al. [22] fused five different CBMs (of families CBM2, 3, 11, and 30) to the C-terminus of cis-epoxysuccinic acid hydrolase (CESH) and found a 5-times higher half-life for the CBM30-CESH than of wild-type CESH at 30 °C.

### 2.4. Adsorption and Enzyme Kinetic Parameters

The binding capacity to WMS granules was increased for all three SBD-TuαGT fusions, revealing that the SBD domains were functional and fulfilling the purpose (Figure 4). Overall, SBD_GA_-TuαGT had an almost 5 times higher binding capacity (B_max_, Figure 4) and 10 times stronger affinity (*K*_d_ = 0.7 mg/mL) than TuαGT (*K*_d_ = 7.2 mg/mL). While SBD_St1_-TuαGT and SBD_St2_-TuαGT both had an essentially 3 times higher binding capacity to WMS granules than TuαGT, their affinity was quite similar and 5-fold larger, respectively, than of TuαGT (Figure 4). This agrees with SBD_St1_ lacking the lysine (Lys578, *An*GA numbering) and SBD_St2_ missing one of the two tryptophans (Trp590, *An*GA numbering) at starch binding site one, respectively, compared with SBD_GA_ (see Section 2.2; Appendix A). Notably, the positive effect of SBD_St2_ on binding was larger than of SBD_St1_ even though SBD_St2_ misses a tryptophan at binding site one, indicating that other features of these SBDs contribute to their binding determinants for WMS granules. This may likely include differences at the larger and more flexible binding site two, which is claimed for SBD_GA_ to be the tighter binding of the two sites [8,9]. Until now, there has been no report of different functions of the two SBDs arranged in tandem in *St*DPE2 or in other DPE2 enzymes.

The fusion of SBDs to TuαGT also influenced the enzymatic activity. Thus, the maltotriose disproportionation was reduced, SBD_St2_-TuαGT and SBD_GA_-TuαGT having slightly lower *K*_m_ than TuαGT, but 4-fold lower *k*_cat_, and yielding 3-fold lower catalytic efficiency (*k*_cat_/*K*_m_) for these two fusion enzymes. Notably, *k*_cat_/*K*_m_ for SBD_St1_-TuαGT was 15-times reduced compared with TuαGT, due to a doubled *K*_m_ and an almost 9-fold lower *k*_cat_ (Table 2). By contrast, using amylose as a substrate, the SBD-fusion improved activity and kinetic parameters somewhat (Table 2). Thus, the similar *K*_m_ and higher *k*_cat_ of SBD_St2_-TuαGT more than doubled the catalytic efficiency compared with TuαGT, whereas the overall outcome for SBD_St1_-TuαGT and SBD_GA_-TuαGT was essentially the same catalytic efficiency as of the parent enzyme. Overall, the kinetic analyses indicated that the SBD-fusion hampered the action of TuαGT on the oligosaccharide (maltotriose), but could improve it on the polysaccharide (amylose). Similarly, fusion of the SBD_GA_ to barley α-amylase, albeit via the much longer natural linker from *A. niger* glucoamylase (*An*GA), showed no adverse effect of the SBD on the active site integrity, as it did not change activity for soluble starch [23]. The improved catalytic efficiency for SBD_St2_-TuαGT towards amylose may be caused by favourable polysaccharide binding to SBD_St2_, increasing the local substrate concentration and perhaps also directing the substrate chain to the active site on the CD.

### 2.5. Hydrolysis and Cyclization Activities on Different Substrates

To gain insight into the modes of action of the SBD-TuαGT fusions on starch, the hydrolysis and cyclization activities were determined using different substrates (Table 3). SBD_St1_-TuαGT had 1.3–1.7-fold higher hydrolytic activity on amylose and gelatinised starch and 1.5-fold higher cyclization activity on amylose than the TuαGT parent enzyme. Similarly, SBD_St2_-TuαGT showed 1.5–1.7-fold increased hydrolysis of gelatinised starch, but more moderate 1.3-fold and 1.2-fold increased hydrolytic and cyclization activities, respectively, on amylose. As a glucanotransferase, it is not expected to show increased hydrolysis by SBD-fusion. However, from an industrial viewpoint, a small increase in hydrolytic activity can help to decrease the viscosity of gelatinised starch, which will also facilitate the TuαGT disproportionation reaction. Notably, for SBD_GA_-TuαGT containing an SBD that originates from the family GH15 of glucoamylases and not from the family GH77 of 4-α-glucanotransferases, to which TuαGT belongs, the hydrolysis and cyclization activities were both essentially the same as for the parent enzyme, except for a slight increase in hydrolysis of gelatinised waxy maize starch (WMS) (Table 3). We speculate that, perhaps, the domain architecture matters and the naturally N-terminally placed SBDs from the *St*DPE2 of the family GH77, which constitutes glycoside hydrolase clan H together with GH13 and GH70 [1], are able to provide support in the different GH77 4-α-glucanotransferase reactions as opposed to the naturally C-terminally placed SBD_GA_ connected via a long *O*-glycosylated linker to the CD of glucoamylase of the family GH15 that acts in an *exo*-manner on non-reducing ends of malto-oligosaccharides and α-glucans catalysing release of glucose [24].

### 2.6. Structure Analysis of Modified NMS

The modification of maize starch both by TuαGT and the SBD-TuαGT fusions significantly affected its structural properties. Chain length distribution (CLD) of NMS and modified NMS (Figure 5A) and the percentage of A-chains as well as of B_1_-, B_2_-, and B_3_-chains (Table 4) showed that all NMS starches treated by TuαGT and its SBD-fusions, to different degrees, contained significantly fewer of the short A-chains and more of the longer B_1_-, B_2_-, and B_3_-chains. Still, only minor differences appeared for the CLD in starches modified by the TuαGT parent compared with SBD-TuαGT fusions (Figure 5A). Previous studies on tapioca starch similarly indicated that exterior chains of amylopectin were elongated by TuαGT [13].

The molecular weight distribution of NMS before and after enzyme treatment was analysed by SEC-MALLS-RI (Figure 5B). Before modification, typical amylopectin (peak 1) and amylose (peak 2) molecules were observed in NMS by SEC. However, after the enzyme modification, three peaks were observed, namely the peaks one and two as well as a distinct later eluting peak three of smaller polysaccharide chains. Furthermore, a later elution of peak one from all modified starch samples indicated that amylopectin has a reduced molecular weight and was less well resolved from peak two than found for unmodified NMS. The newly appearing prominent peak three of smaller molecules may contain large-ring cyclodextrins (LR-CDs) produced in cyclization reactions [25] as well as polysaccharide hydrolysis products.

To further understand the reaction of TuαGT and the SBD-TuαGT fusions, the α-1,6/α-1,4-linkage ratio that indicates the degree of branching, was determined for the modified starches by using ^1^H-NMR (Figure 5C). NMS modified by TuαGT and SBD_GA_-TuαGT showed a slight increase in the α-1,6/α-1,4-linkage ratio from 3.76 for unmodified to 3.84 and 3.88%, respectively, after modification, whereas treatment by SBD_St1_-TuαGT and SBD_St2_-TuαGT increased the ratio to 4.13 and 4.08%, respectively. As TuαGT can catalyze hydrolysis, disproportionation, cyclization, and coupling, which all involve α-1,4-linkages, the increase in the α-1,6/α-1,4-linkage ratio can reflect the level of hydrolysis, in which α-1,4 linkages are lost and not generated, in agreement with the two fusions with SBD_St1_ and SBD_St2_, i.e., the SBDs from *St*DPE2 belonging to the family GH77, showing an increased degree of hydrolysis of gelatinised NMS compared with TuαGT (Figure 5B; Table 3).

## 3. Material and Methods

### 3.1. Materials

Amylose (potato), maltotriose, and protease inhibitor cocktail tablets (cOmplete™, Mini, EDTA-free Protease Inhibitor Cocktail) were purchased from Sigma-Aldrich Co. Ltd. (St. Louis, MO, USA). Pullulanase M2 (from *Bacillus licheniformis*, 900 U/mL) and β-amylase (from barley, 600 U/mg) were purchased from Megazyme Co. Ltd. (Wicklow, Ireland). Waxy maize starch (WMS) was the kind gift of Cargill (USA) and normal maize starch (NMS) of Archer Daniels Midland (ADM, Decatur, IL, USA).

### 3.2. Bioinformatics Analysis of CBM20

In total, 87 CBM20 domains from 65 different amylolytic and related enzymes were collected (Table 1) based on previous studies focused on GH77 DPE2s and different starch-binding domain CBM families [5,16,17,18,19]. All sequences were retrieved from GenBank (https://www.ncbi.nlm.nih.gov/genbank/, accessed on 23 December 2022; [26]) and/or UniProt (https://www.uniprot.org/, accessed on 23 December 2022) [27]) sequence databases. For DPE2s selected from various bacteria and eukaryotes, the number of CBM20 copies and their borders in respective sequences were taken from UniProt [27] and complemented by data available from the literature [5,16,17,18]; questionable cases were also verified in the InterPro database (https://www.ebi.ac.uk/interpro/, accessed on 23 December 2022 [28]). Although each studied DPE2 could eventually contain two CBM20 copies in tandem at their N-terminus, putative CBM20 copies that lacked most of the functionally important binding site residues were not considered (Table 1). For CAZymes, the appropriate CAZy classification has been checked against the CAZy database (http://www.cazy.org/, accessed on 23 December 2022; [1]) and published data [5,16,17,18,19]. Sequences were aligned using the program Clustal Omega (https://www.ebi.ac.uk/Tools/msa/clustalo/, accessed on 23 December 2022; [29]) and the alignment was confirmed by comparison of three-dimensional structures of selected CBM20s: (i) two experimentally determined structures from *Aspergillus niger* GH15 glucoamylase [8,9] and *Bacillus circulans* GH13_2 cyclodextrin glucanotransferase [19] retrieved from Protein Data Bank (PDB; https://www.rcsb.org/, accessed on 23 December 2022; [30]) under their PDB codes 1AC0 and 1CXE, respectively; and (ii) the modelled structure of *Solanum tuberosum* GH77 DPE2 taken from the AlphaFold database (https://alphafold.ebi.ac.uk, accessed on 23 December 2022; [31]) via its UniProt accession No.: Q6R608. The corresponding CBM20 structures were superimposed using the program MultiProt (http://bioinfo3d.cs.tau.ac.il/MultiProt/, accessed on 23 December 2022; [32]). Since the structure superimpositions did not identify any significant discrepancies with the sequence alignment, the Clustal Omega program-produced output was used for calculating the maximum-likelihood evolutionary tree by the bootstrapping procedure with 1000 bootstrap trials [33], implemented in the MEGA-X package [34]. The calculated tree file was displayed with the program iTOL (https://itol.embl.de/, accessed on 23 December 2022; [35]).

### 3.3. Construction of TuαGT and SBD-TuαGT Fusions

4-α-Glucanotransferase from *Thermoproteus uzoniensis* (TuαGT, GenBank Accession WP_013679179.1) was produced recombinantly essentially as described [13]. Genes codon-optimised for *Escherichia coli* encoding full-length TuαGT connected N-terminally to the indicated SBD (SBD_St1_, Uniprot Accession Q6R608_2 residues 3–112; SBD_St2_, Uniprot Accession Q6R608_2 residues 147–259; SBD_GA_, Uniprot Accession P69328.1, residues 538–639) via an 18-residues linker (TTGESRFVVLSDGLMREM), that naturally connects the SBD_St1_-SBD_St2_ tandem with the CD in *St*DPE2 [14], were purchased and cloned into the expression vector pET-28a (+) using the restriction sites NheI and XhoI (GenScript, Leiden, The Netherlands) in frame with the N-terminal His-tag.

### 3.4. Production of TuαGT and SBD-TuαGT Fusions

TuαGT, SBD_St1_-TuαGT, SBD_St2_-TuαGT, and SBD_GA_-TuαGT encoding plasmids were transformed into *E. coli* BL21(DE3)* and screened on Lysogeny broth (LB) agar containing 50 µg/mL kanamycin for selection. Starter cultures (10 mL) made by inoculating LB medium (1% tryptone, 0.5% yeast extract, 0.5% NaCl, 50 µg/mL kanamycin) with a single colony and incubating (37 °C, 170 rpm, overnight) were used to inoculate 800 mL LB medium containing 10 mM glucose and 50 μg/mL kanamycin in shake flasks. Expression was induced at A_600_ = 0.6 by adding isopropyl-β-D-thiogalactopyranoside (IPTG) to 0.2 mM and incubated (18 °C, 160 rpm, 24 h). The cells were harvested (4000× *g*, 4 °C, 30 min) and stored at −20 °C until protein purification.

### 3.5. Purification of TuαGT and SBD-TuαGT Fusions

Cells (5 g) were thawed and resuspended in 20 mL HisTrap equilibration buffer (20 mM Hepes, 250 mM NaCl, 10% glycerol, pH 7.5), added 1 protease inhibitor cocktail tablet, lysed using a high-pressure homogenizer at 1 bar, added 2 μL Benzonase Nuclease (Sigma-Aldrich, St. Louis, MO, USA), and centrifuged (40,000× *g*, 4 °C, 30 min). The supernatant (~20 mL) was mixed with 2 mL HisPur^TM^ nickel-nitrilotriacetic acid resin (Thermo Fisher Scientific, Waltham, MA, USA) pre-equilibrated with equilibration buffer and washed with 20 column volumes (CV) of equilibration buffer, added 10 mM imidazole. Bound protein was eluted by 10 CV of equilibration buffer, added 300 mM imidazole. Protein-containing fractions were pooled (10 mL) and further purified by gel filtration (Superdex 16/60 200 pre-equilibrated with 20 mM Hepes, 150 mM NaCl, 10% glycerol, pH 7.5) at a flow rate of 1 mL/min. Fractions containing disproportionation activity on maltotriose were pooled and buffer-exchanged to ion exchange chromatography equilibration buffer (20 mM Hepes, 10% glycerol, pH 7.5) using Amicon^®^ Ultra-15 Centrifugal Filter Unit (Ultracel-30 regenerated cellulose membrane, 15 mL sample volume, Merck), concentrated to 2 mL using centrifugal filters (30 kDa MWCO; Amicon^®^ Ultra), filtrated (0.45 μm), and loaded onto a Resource Q column (1 mL, Cytiva), pre-equilibrated with 15 CV equilibration buffer, and eluted by 50 CV of a linear gradient from 0 to 800 mM NaCl in equilibration buffer. Fractions presenting activity were verified by SDS-PAGE to contain TuαGT, SBD_St1_-TuαGT, SBD_St2_-TuαGT, and SBD_GA_-TuαGT with theoretical molecular weights calculated to 55,593, 68,272, 67,068, and 69,562 Da, respectively (https://web.expasy.org/protparam/, accessed on 23 December 2022). Protein concentrations were determined spectrophotometrically at 280 nm (Nanodrop Lite, Thermo Scientific, USA) using theoretical extinction coefficients (ε) for TuαGT, SBD_St1_-TuαGT, SBD_St2_-TuαGT and SBD_GA_-TuαGT of 141,750, 172,690, 160,200, 172,630 M^−1^cm^−1^, respectively (https://web.expasy.org/protparam/, accessed on 23 December 2022). Recombinant SBD-TuαGT fusion proteins and TuαGT wild type were obtained in yields of 0.05–0.1 and 2.5 mg, respectively, per 5 g *E. coli* cells from 0.8 L culture.

### 3.6. Enzyme Activity Assays

#### 3.6.1. Total Activity

The total activity of TuαGT and the SBD-TuαGT fusions was determined by incubating amylose (2 mg/mL) in 900 μL assay buffer (50 mM Hepes, pH 7.0, 150 mM NaCl) with 100 μL enzyme (20 nM, final concentration) at 75 °C for 10 min [13]. The reaction was terminated by heating (99 °C, 15 min), and the amylose concentration was determined by mixing 20 μL heated sample with 200 μL iodine reagent (0.2% KI + 0.02% I_2_) for 1 min. The absorbance was measured at 620 nm (microplate reader, PowerWave XS, BIO-TEK) [36]. One unit of total activity was defined as the amount of enzyme degrading 0.5 mg/mL amylose per min under the above conditions.

#### 3.6.2. Disproportionation

The disproportionation activity of TuαGT and SBD-TuαGT fusions was determined as reported [13] by incubating 1% (19.8 mM) maltotriose in 900 μL assay buffer (see Section 3.6.1) with 100 μL enzyme (10 nM, final concentration) at 75 °C for 1 h. The reaction was terminated (99 °C, 15 min) and the released glucose was quantified using the GOPOD assay (D-Glucose Assay Kit, Megazyme) with glucose (0–1000 μM) as standard [37]. One unit of disproportionation activity was defined as the amount of enzyme releasing 1 μmol/min glucose under the above conditions.

#### 3.6.3. Hydrolysis

The hydrolytic activity of TuαGT and the SBD-TuαGT fusions was determined by incubating 2 mg/mL amylose in 900 μL assay buffer (see Section 3.6.1) with 100 μL enzyme (20 μM, final concentration) at 70 °C for 1 h [38]. Hydrolytic activity towards 25 mg/mL NMS (gelatinised at 99 °C, 30 min, 1100 rpm, and cooled to 70 °C before the assay) was determined by addition of enzyme (2 μM, final concentration) and incubated (70 °C, 1 h). The reaction was stopped by the PAHBAH reagent (1:1, *v*:*v*), heating (95 °C, 10 min) [39] and the absorbance was measured at 405 nm after cooling. One unit of activity was defined as the amount of enzyme releasing 1 μmol/min reducing sugar under the above conditions. Glucose (0–1000 μM) was used for the standard curve.

#### 3.6.4. Cyclization

The cyclization activity of TuαGT and SBD-TuαGT fusions was determined by incubating 2 mg/mL amylose in 900 μL assay buffer (see Section 3.6.1) with 100 μL enzyme (20 μM, final concentration) at 70 °C for 1 h [40]. The reaction was terminated (99 °C, 15 min), and 0.24 U β-amylase was added and incubated at 40 °C for 10 h to degrade remaining amylose. The reaction was stopped by adding the PAHBAH reagent (1:1, *v*:*v*) and the absorbance was measured at 405 nm (as in Section 3.6.3). The amount of formed cycloamylose was determined by the difference of maltose released by β-amylase from untreated amylose and from amylose treated with TuαGT and SBD-TuαGT fusions. One unit of cyclization activity was defined as the amount of enzyme leading to release of 1 μmol less maltose per min under the above conditions using maltose (0–1000 μM) for the standard curve.

### 3.7. Effect of pH and Temperature on Activity

The pH optimum was determined at the optimum temperature 70 °C of TuαGT using the disproportionation activity assay (see Section 3.6.2) in universal buffer (20 mM MES, 20 mM Hepes, 150 mM NaCl, pH 4.0–9.0) [41]. The temperature optimum in the range of 50–90 °C was determined at the optimum pH 7.0 of TuαGT in the above buffer. To assess thermostability, TuαGT and SBD-TuαGT fusions (100 nM) were incubated at 50 °C and pH 7.0 (50 mM Hepes buffer, 150 mM NaCl) and the residual enzyme activity was measured during 8 h with 1 h intervals. The activity before incubation defined 100% stability.

### 3.8. Kinetic Parameters

Enzyme (10 nM, final concentration) was incubated (70 °C, 300 rpm) with maltotriose (1 mL; six concentrations, 0.5–7.5 μM) in assay buffer (see Section 3.6.1). Aliquots (100 μL) removed at 1, 2, 5, 10, 15 min were mixed with 20 μL 0.2 M NaOH (10 min), neutralized by 20 μL 0.2 M HCl, and the rate of glucose release was determined (see Section 3.6.2). Enzyme (10 nM, final concentration) was incubated (70 °C, 300 rpm) with amylose (1 mL; six concentrations, 0.1–2 mg/mL) in assay buffer (see Section 3.6.1). Aliquots (100 μL) removed at 1, 2, 5, 10, 15 min were mixed with DNS reagent (100 μL) and heated (99 °C, 5 min). After cooling, the absorbance was measured at 520 nm. *V*_max_, *K*_m_, and *k*_cat_ were calculated by fitting the Michaelis–Menten equation using GraphPad Prism 6 (GraphPad Software Inc., San Diego, CA, USA).

### 3.9. Adsorption to Starch Granules

The binding capacity of TuαGT and SBD-TuαGT fusions on WMS granules at 25 °C was determined under the same conditions as used for the activity assay (see Section 3.6.1) by adding enzyme (200 nM, final concentration) to different WMS concentrations from 0.5 to 75 mg/mL [42]. After 10 min the mixtures were centrifuged (10,000× *g*, 5 min) and 100 μL supernatant was added to 100 μL 2.5-fold diluted protein assay dye reagent (Bio-Rad). The enzyme concentration was determined from the ratio of absorbance values at 590 over 450 nm using TuαGT and SBD-TuαGT (0–1.0 μM) as standards. The Langmuir isotherm (Equation (1)) is a commonly used model for analysis of molecular binding and was fitted to the results using GraphPad Prism 6 (GraphPad Software Inc.), where K_d_ is the dissociation constant, Γ is the bound protein concentration, and B_max_ is the (apparent) saturation coverage.
(1)$$\Gamma =\frac{B_{\max}\cdot E_{\mathrm{free}}}{K_{d}+E_{\mathrm{free}}}$$

### 3.10. Preparation of Modified Maize Starch (MMS)

Enzymatic modification of NMS was performed essentially as reported [13]. Starch (6%, *w*/*v*) was suspended in activity assay buffer (see Section 3.6.1) and gelatinised (99 °C, 30 min, 1100 rpm). The modification was carried out by 1 μmol TuαGT or SBD-TuαGT fusions per 1 g starch at 70 °C for 8 h, and terminated by heating (99 °C, 30 min). The modified starch was precipitated by three volumes of ethanol overnight and isolated by centrifugation (4000× *g*, 10 min). The precipitated starch was kept overnight at −80 °C and freeze-dried for further analysis.

### 3.11. Molecular Weight Distribution

Size exclusion chromatography with multi-angle laser light scattering-refractive index detector (SEC-MALLS-RI) was used to analyse the molecular weight of starch samples [43]. Dry starch (5 mg/mL) was suspended in a mixture of DMSO and MilliQ water (9:1, *v*/*v*) and gelatinised on a boiling water bath (1 h, shaking every 10 min) until the solution was clear and free of floc. The gelatinised starch was incubated (30 °C, 250 rpm, 48 h) to disrupt remaining starch particles. The samples were re-boiled and filtrated through a 0.45 μm filter. Filtrate (100 µL) was injected on a tandem column (Ohpak SB-804 HQ, Ohpak SB-806 HQ) using 0.1 M NaNO_3_ (in 0.02% NaN_3_) as mobile phase at a flow rate of 0.6 mL/min with the column temperature set at 50 °C. Data obtained from the MALLS and RI detectors were analysed by ASTRA software version 5.3.4 (Wyatt Technologies).

### 3.12. Chain Length Distribution

High performance anion exchange chromatography with pulsed amperometric detection (HPAEC-PAD) was used to analyse the chain length distribution of NMS before and after enzyme modification. Starch (5 mg/mL, dry solid (*w*/*v*)) was suspended in 50 mM sodium acetate, pH 4.5, followed by gelatinisation (99 °C, 30 min). The gelatinised starch was debranched by incubation with 0.18 U pullulanase per 5 mg starch at 42 °C for 12 h and centrifuged (10,000× *g*, 10 min). The supernatant was analysed by HPAEC-PAD [44].

### 3.13. 1H-NMR

1D ^1^H NMR spectra of starch samples were acquired using a 600 MHz NMR spectrometer (Bruker Avance III, Bruker Biospin, Rheinstetten, Germany) [45]. Starch (5 mg/mL, dry solid (*w*/*v*)) was suspended in D_2_O, gelatinised (99 °C, 2 h), freeze-dried twice, dissolved in DMSO-d6 (90% DMSO-d6 in 10% D_2_O), and heated (99 °C, 30 min) before analysis. The percentage of glucan branch points of starch samples was estimated using the areas of signals representing anomeric protons (δ 5.35–5.45 α-1,4; δ 4.95–5.00 α-1,6).

## 4. Conclusions

In the present work, three phylogenetically diverse SBDs, two from *St*DPE2 and one from *An*GA, fused one by one via an 18-residues linker to the N-terminus of the thermophilic 4-α-glucotransferase (TuαGT), conferred the TuαGT with altered distinct substrate binding and activity characteristics. The bioinformatics analysis shows the distant relationship between SBD_St1_, SBD_St2_, and SBD_GA_ each found in well-separated clusters of the evolutionary tree and sharing this position with close homologues, i.e., copies one and two of GH77 DPE2s and SBDs from various CAZymes. Relative to the parent enzyme TuαGT, the SBD_St2_-fusion had improved thermostability after 5 h of thermal treatment and also doubled the disproportionation activity on amylose. By contrast, all three SBD-fusions decreased the disproportionation activity using maltotriose as substrate. The SBD_GA_-fusion resulted in the highest binding affinity and binding capacity on starch granules, presumably reflecting the superior function of the two binding sites in this SBD containing all of the canonical aromatic residues. The structural analysis of starch before and after modification by TuαGT and the three SBD-fusion enzymes indicated that the fusion with SBD_St1_ and SBD_St2_ enhanced hydrolysis the most, along with their highest cyclization activity, and a slightly higher loss of the short A chains and gain of B chains, which is caused by the disproportionation reaction, compared with fusion by SBD_GA_. As is known for TuαGT, the starch products may represent nutritional values reminiscent of resistant starch dietary fibres. According to the separation in the evolutionary tree and the different functional improvements, we conclude that SBD_St1_ and SBD_St2_ contribute different effects by fusion with TuαGT and that they probably play different, albeit not yet identified, functional roles in the *St*DPE2. In the longer perspective, the obtained results disclose the potential for utilising insight into the wide diversity of SBDs for enzyme engineering and also to connect individual properties of the two “in tandem” SBDs with structure/function relationships of disproportionating enzymes in plants and bacteria.

## Figures and Tables

**Figure 1 molecules-28-01320-f001:**
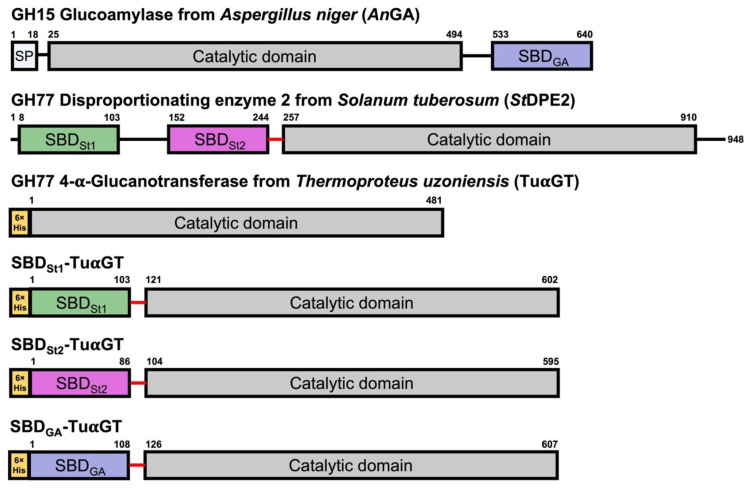
Domain architecture of amylolytic enzymes used in the present study. *Aspergillus niger* glucoamylase (*An*GA), *Solanum tuberosum* disproportionating enzyme 2 (*St*DPE2), 4-α-glucanotransferase from *Thermoproteus uzoniensis* (TuαGT), and the three SBD-TuαGT fusions (SBD_St1_-TuαGT, SBD_St2_-TuαGT, and SBD_GA_-TuαGT) containing full length TuαGT and an SBD of family CBM20 connected to the N-terminus via an 18-residues linker (red: TTGESRFVVLSDGLMREM).

**Figure 2 molecules-28-01320-f002:**
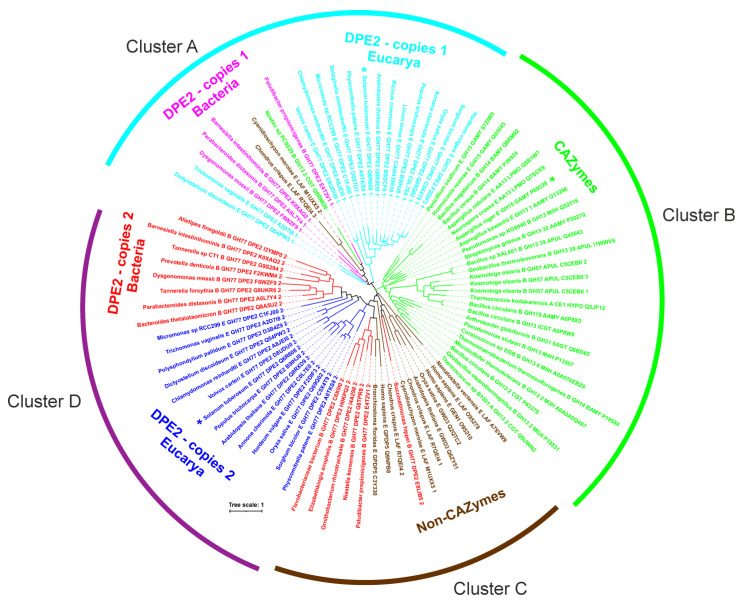
Phylogenetic tree of CBM20s with focus on GH77 DPE2s. The tree is based on the alignment of entire CBM20 sequences (Appendix A). The labels of protein sources consist of the name of the organism, letter “A”, “B”, or “E” for the archaeal, bacterial, and eukaryotic origin, respectively, CAZy family affiliation (if any), enzyme abbreviated name (for details, see Table 1), and the UniProt accession number. If there are more CBM20 copies in a single protein, the copies in the order of their appearance in the sequence are also indicated by the relevant number “1”, “2”, and “3” (at the end of the protein label). The three CBM20 domains, two from GH77 *Solanum tuberosum* DPE2 and one from GH15 *Aspergillus niger* glucoamylase, studied in the present work, are marked by an asterisk.

**Figure 3 molecules-28-01320-f003:**
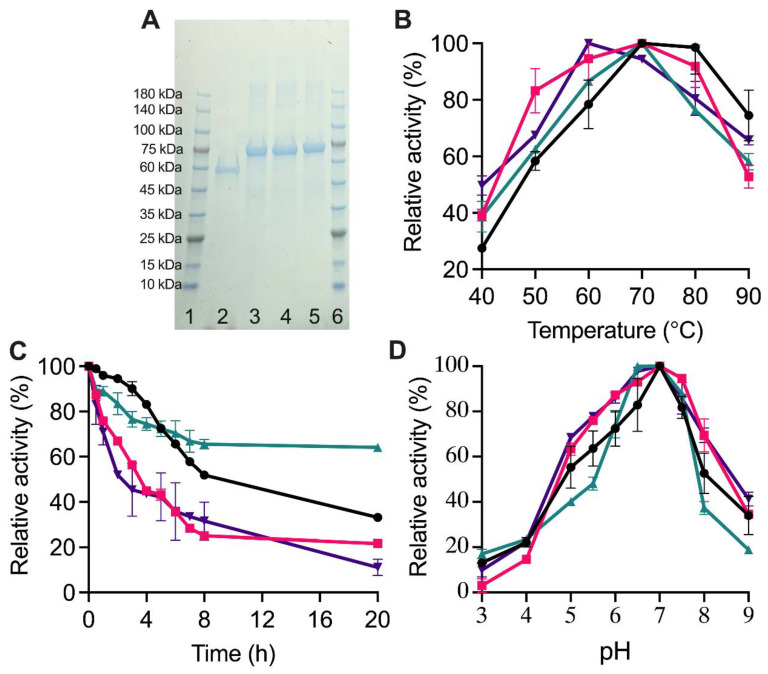
Biochemical characterization of TuαGT and SBD-TuαGT fusions. (**A**) SDS-PAGE of purified enzymes. Lanes 1 and 6: Marker, Lane 2: TuαGT (6.5 μg), Lane 3: SBD_St1_-TuαGT (6.5 μg), Lane 4: SBD_St2_-TuαGT (6.5 μg), Lane 5: SBD_GA_-TuαGT (6.5 μg); (**B**) Temperature dependence for maltotriose disproportionation; (**C**) Thermostability at 50 °C; (**D**) pH dependence for maltotriose disproportionation. TuαGT (black), SBD_St1_-TuαGT (red), SBD_St2_-TuαGT (green), and SBD_Ga_-TuαGT (purple). Activity at pH or temperature optima was defined as 100% for the individual enzymes.

**Figure 4 molecules-28-01320-f004:**
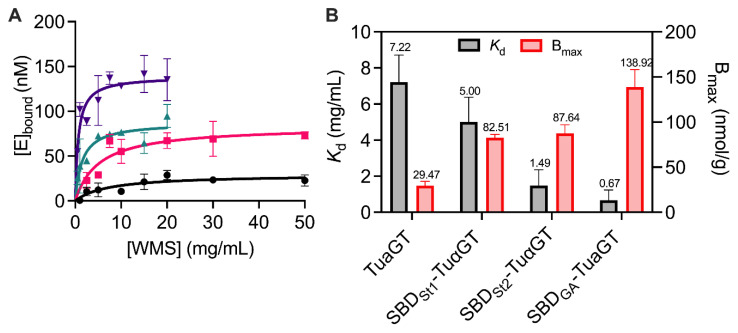
Binding capacity of TuαGT and SBD-TuαGT fusions on waxy maize starch (WMS) granules. (**A**) Binding isotherms on WMS granules for TuαGT (black), SBD_St1_-TuαGT (red), SBD_St2_-TuαGT (green), and SBD_GA_-TuαGT (purple) at 25 °C and pH 7.0. Lines represent best fits of the Langmuir adsorption isotherm. (**B**) Dissociation constant (*K*_d_) and (apparent) saturation coverage (B_max_) on WMS granules.

**Figure 5 molecules-28-01320-f005:**
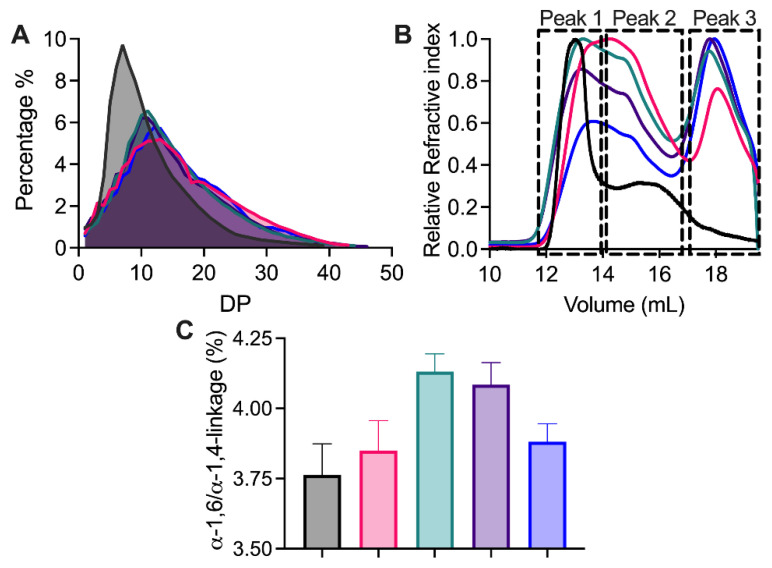
Structural analysis of NMS modified by TuαGT and SBD-TuαGT fusions. (**A**) Chain length distribution; (**B**) Molecular weight distribution; (**C**) ^1^H-NMR analysis of α-1,6/α-1,4 linkage ratio. Before (black), after modification by TuαGT (red), SBD_St1_-TuαGT (green), SBD_St2_-TuαGT (purple), and SBD_GA_-TuαGT (blue).

**Table 1 molecules-28-01320-t001:** The CBM20s originating from DPE2s, various other CAZymes, and related enzymes used in the present study *^a^*.

No.	B/A/E *^b^*	Organism	Family *^c^*	Enzyme *^d^*	GenBank *^e^*	UniProt *^e^*	Length *^f^*	CBM20_1 *^g^*	CBM20_2 *^g^*	CBM20_3 *^g^*	Insert *^h^*
1	E	*Annona cherimola*	GH77	DPE2	ACN50178.1	C0L7E0	953	10–119	154–268		606–750
2	E	*Arabidopsis thaliana*	GH77	DPE2	AAL91204.1	Q8RXD9	955	13–122	157–270		608–752
3	E	*Chlamydomonas reinhardtii*	GH77	DPE2	EDO97689.1	A8JEI0	941	1–119	155–271		631–775
4	E	*Dictyostelium discoideum*	GH77	DPE2	EAL65318.1	Q54PW3	907	1–102	134–241		594–729
5	E	*Hordeum vulgare*	GH77	DPE2	BAJ94874.1	F2DIF3	931	1–108	143–257		595–739
6	E	*Linum tenue*	GH77	DPE2	CAI0439830.1	---	1137	10–119			499–643
7	E	*Micromonas* sp. RCC299	GH77	DPE2	ACO70268.1	C1FJ00	975	1–114	169–286		636–793
8	E	*Oryza sativa*	GH77	DPE2	BAD31425.1	Q69Q02	946	7–115	150–264		602–746
9	E	*Physcomitrella patens*	GH77	DPE2	EDQ55980.1	A9TKS8	1006	14–123	165–279		618–763
10	E	*Polysphondylium pallidum*	GH77	DPE2	EFA84397.1	D3B4Z9	1070		167–279		627–761
11	E	*Populus trichocarpa*	GH77	DPE2	EEF04969.1	B9IHJ8	975	10–119	155–268		606–750
12	E	*Ricinus communis*	GH77	DPE2	EEF38704.1	B9SCF0	901	10–119			533–676
13	E	*Selaginella moellendorffii*	GH77	DPE2	EFJ19739.1	D8S7D7	930	15–128			600–740
14	E	*Solanum tuberosum*	GH77	DPE2	AAR99599.1	Q6R608	948	1–112	147–259		597–741
15	E	*Sorghum bicolor*	GH77	DPE2	EER97686.1	C5X4T9	946	6–114	149–263		601–745
16	E	*Trichomonas vaginalis*	GH77	DPE2	EAY23705.1	A2D7I8	930	1–112	142–249		594–704
17	E	*Volvox carteri*	GH77	DPE2	EFJ42152.1	D8UDU0	995	51–178	214–329		671–786
18	B	*Alistipes finegoldii*	GH77	DPE2	AFL78258.1	I3YMP0	867		115–225		556–691
19	B	*Bacteroides thetaiotaomicron*	GH77	DPE2	AAO77253.1	Q8A5U2	893		119–235		573–714
20	B	*Barnesiella intestinihominis*	GH77	DPE2	EJZ64889.1	K0XAQ2	893	1–97	123–239		577–718
21	B	* Dysgonomonas mossii *	GH77	DPE2	EGK04046.1	F8WZF9	888	1–95	119–231		571–712
22	B	*Elizabethkingia anophelis*	GH77	DPE2	EHM98897.1	H0KPQ2	885		119–225		572–711
23	B	*Flavobacteriaceae bacterium*	GH77	DPE2	ACU06866.1	C6X0I0	884		117–226		570–709
24	B	*Niastella koreensis*	GH77	DPE2	AEV98902.1	G8TPR9	895		127–241		579–720
25	B	*Ornithobacterium rhinotracheale*	GH77	DPE2	AFL98082.1	I4A298	874		109–217		563–698
26	B	*Paludibacter propionicigenes*	GH77	DPE2	ADQ79045.1	E4T2V1	897	1–101	128–243		582–722
27	B	*Parabacteroides distasonis*	GH77	DPE2	ABR41798.1	A6L7Y4	895	1–98	124–240		578–719
28	B	*Prevotella denticola*	GH77	DPE2	AEA21596.1	F2KWM4	897		126–233		581–722
29	B	*Succinatimonas hippei*	GH77	DPE2	EFY07743.1	E8LIB5	879		112–223		562–703
30	B	*Tannerella forsythia*	GH77	DPE2	AEW22695.1	G8UKR6	881		108–223		561–701
31	B	*Tannerella* sp. CT1	GH77	DPE2	EHL87887.1	G9S294	894		124–232		577–718
32	E	*Aspregillus kawachii*	GH13_1	AAMY	BAA22993.1	O13296	640	533–640			
33	B	*Bacillus circulans*	GH13_2	CGT	CAA55023.1	P43379	713	608–713			
34	B	*Geobacillus stearothermophilus*	GH13_2	MGA	AAA22233.1	P19531	719	609–719			
35	B	*Nostoc* sp. PC9229	GH13_2	CGT	AAM16154.1	Q8RMG0	642	534–642			
36	B	*Microbulbifer thermotolerans*	GH13_2	M3H	AID53183.1	A0A0A0Q4S7	761	657–761			
37	A	*Thermococcus* sp. B1001	GH13_2	CGT	BAA88217.1	Q9UWN2	739	629–739			
38	B	*Coralococcus* sp. EGB	GH13_6	M6H	AII00648.1	A0A076EBZ6	522	421–522			
39	B	*Streptomyces griseus*	GH13_32	AAMY	CAA40798.1	P30270	566	465–566			
40	B	*Geobacillus thermoleovorans*	GH13_39	APUL	AFI70750.1	I1WWV6	1655	1252–1349			
41	B	*Bacillus* sp. XAL601	GH13_39	APUL	BAA05832.1	Q45643	2032	1330–1427			
42	B	*Pseudomonas stutzeri*	GH13	M4H	AAA25707.1	P13507	548	446–548			
43	B	*Pseudomonas* sp. KO-8940	GH13	M5H	BAA01600.1	Q52516	614	509–614			
44	B	*Bacillus circulans*	GH13	ICGT	BAF37283.1	A0P8W9	995	888–995			
45	B	*Bacillus cereus*	GH14	BAMY	BAA75890.1	P36924	551	444–551			
46	B	*Bacillus megaterium*	GH14	BAMY	CAB61483.1	Q9RM92	545	444–545			
47	B	*Thermoanaerobacterium thermosulfurogenes*	GH14	BAMY	AAA23204.1	P19584	551	448–551			
48	E	*Aspergillus niger*	GH15	GAMY	CAA25303.1	P69328	640	533–640			
49	E	*Hormoconis resinae*	GH15	GAMY	CAA48243.1	Q03045	616	501–608			
50	E	*Penicillium oxalicum*	GH15	GAMY	EPS30575.1	S7ZIW0	616	508–616			
51	B	*Arthrobacter globiformis*	GH31	6AGT	BAD34980.1	Q6BD65	965	859–965			
52	B	*Kosmotoga_olearia*	GH57	APUL	ACR80150.1	C5CEB0	1354	32–136	155–258	267–372	
53	B	*Bacillus circulans*	GH119	AAMY	BAF37284.1	A0P8X0	1290	1183–1290			
54	E	*Aspergillus nidulans*	AA13	LPMO	CBF81866.1	Q5B1W7	385	278–385			
55	E	*Neurospora crassa*	AA13	LPMO	EAA34371.2	Q7SCE9	385	278–385			
56	A	*Thermococcus kodakarensis*	CE1	HYPO	BAD84711.1	Q5JF12	449	83–188			
57	E	*Arabidopsis thaliana*		GWD3	AAC26245.1	Q6ZY51	1196	66–166			
58	E	*Oryza sativa*		GWD3	ABA97816.2	Q2QTC2	1206	67–168			
59	E	*Branchiostoma floridae*		GPDP5	EEN65442.1	C3Y330	680	1–110			
60	E	*Homo sapiens*		GPDP5	BAA92672.1	Q9NPB8	672	1–115			
61	E	*Homo sapiens*		GEN1	AAC78827.1	O95210	358	258–358			
62	E	*Chondrus crispus*		LAF	CDF36183.1	R7QEI4	549	1–100	167–282	285–387	
63	E	*Cyanidioschyzon merolae*		LAF	BAM83396.1	M1UXX5	532	156–267	268–374		
64	E	*Homo sapiens*		LAF	AAG18377.1	O95278	331	1–124			
65	E	*Nematostella vectensis*		LAF	EDO32135.1	A7SVW9	324	1–125			

*^a^* Sixty-five enzyme sources resulting in eighty-seven CBM20 domains were included in the present study: (i) 17 GH77 DPE2s from *Eukarya* (numbers 1–17)—30 CBM20 sequences; (ii) 14 GH77 DPE2s from *Bacteria* (numbers 18–31)—18 CBM20 sequences; (iii) 25 enzymes representing various other CAZymes (especially amylolytic enzymes; numbers 32–56)—27 CBM20 sequences; and 9 non-CAZymes recognised as possessing CBM20 (numbers 57–65)—12 CBM20 sequences. *^b^* Bacterial (B), archaeal (A), or eukaryotic (E) origin. *^c^* CAZy family/subfamily (if known). *^d^* The abbreviations of enzymes are as follows: DPE2, disproportionating enzyme 2; AAMY, α-amylase; CGT, cyclodextrin glucanotransferase; MGA, maltogenic amylase; M3H, maltotriohydrolase; M6H, maltohexaohydrolase; APUL, amylopullulanase; M4H, maltotetraohydrolase; M5H, maltopentaohydrolase; ICGT, isocyclomaltooligosaccharide glucanotransferase; BAMY, β-amylase; GAMY, glucoamylase; 6AGT, 6-α-glucanotransferase; LPMO, lytic polysaccharide monooxygenase; HYPO, hypothetical protein; GWD3, glucan, water dikinase 3; GPDP5, glycerophosphodiester phosphodiesterase-5; GEN1, genethonin-1; LAF, laforin. *^e^* The Accession Nos. from the GenBank and UniProt databases. *^f^* The length of the protein, i.e., the number of amino acid residues. *^g^* The individual CBM20 copies. *^h^* The insert in DPE2 sequences. The individual groups are distinguished from each other by different colors corresponding to representatives shown in Figure 2 and Appendix A.

**Table 2 molecules-28-01320-t002:** Activity and kinetic parameters of TuαGT and SBD-TuαGT fusions towards maltotriose and amylose at 70 °C and pH 7.0.

Substrate	Parameter	TuaGT	SBD_St1_-TuaGT	SBD_St2_-TuaGT	SBD_GA_-TuaGT
Maltotriose	Activity (U/mg)	27.5 ± 0.7	3.1 ± 0.5	10.3 ± 0.2	7.4 ± 0.4
	*K*_m_ (μM)	1.5 ± 0.1	3.5 ± 0.2	1.1 ± 0.1	1.4 ± 0.1
	*k*_cat_ (s^−1^)	0.04 ± 0.01	0.01 ± 0.0002	0.01 ± 0.002	0.01 ± 0.0005
	*k*_cat_/*K*_m_ (μM^−1^∙s^−1^)	0.03 ± 0.004	0.002 ± 0.0003	0.01 ± 0.001	0.01 ± 0.0004
Amylose	Activity (U/mg)	1.3 ± 0.1	3.1 ± 0.2	2.5 ± 1.1	1.6 ± 0.9
	*K*_m_ (mg/mL)	0.6 ± 0.04	1.9 ± 0.1	0.6 ± 0.1	0.8 ± 0.02
	*k*_cat_ (s^−1^)	2.5 ± 0.3	7.0 ± 0.3	5.5 ± 0.4	3.3 ± 0.2
	*k*_cat_/*K*_m_ (mL∙[mg∙s]^−1^)	3.9 ± 0.2	3.6 ± 0.03	8.6 ± 0.4	4.0 ± 0.2

**Table 3 molecules-28-01320-t003:** Hydrolysis and cyclization by TuαGT and SBD-TuαGT fusions acting on amylose and gelatinised maize starches at 70 °C and pH 7.0.

Activity	Substrate	TuaGT	SBD_St1_-TuaGT	SBD_St2_-TuaGT	SBD_GA_-TuaGT
Cyclization	Amylose	3.2 ± 0.2	4.8 ± 0.2	3.9 ± 0.3	3.3 ± 0.1
Hydrolysis	Amylose	0.3 ± 0.01	0.4 ± 0.01	0.4 ± 0.02	0.3 ± 0.01
	WMS	0.3 ± 0.02	0.5 ± 0.1	0.5 ± 0.1	0.4 ± 0.02
	NMS	0.2 ± 0.02	0.3 ± 0.03	0.3 ± 0.1	0.2 ± 0.1

**Table 4 molecules-28-01320-t004:** Percentage of different chains in normal maize starch (NMS) before and after modification by TuαGT and SBD-TuαGT fusions.

Type of Chain ^a^	NMS	TuaGT	SBD_St1_-TuaGT	SBD_St2_-TuaGT	SBD_GA_-TuaGT
A-chain	67.2 ± 0.4	38.3 ± 0.7	35.8 ± 0.9	40.2 ± 2.0	41.6 ± 0.4
B_1_-chain	28.0 ± 0.7	46.3 ± 2.0	52.0 ± 1.5	43.8 ± 3.0	45.1 ± 2.0
B_2_-chain	4.4 ± 0.2	13.5 ± 0.8	10.2 ± 0.9	11.7 ± 0.9	11.8 ± 1.9
B_3_-chain	0.6 ± 0.03	2.5 ± 0.2	2.5 ± 0.3	2.6 ± 0.6	1.9 ± 0.5

^a^ A-chain: DP 1–12, B_1_-chain: DP 13–24, B_2_-chain: DP 25–36, and B_3_-chains: DP > 37.

## Data Availability

All available data are included in the article.

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
