# Peer review of "Impact of Starch Binding Domain Fusion on Activities and Starch Product Structure of 4-α-Glucanotransferase"

_molecules, 2023, doi:10.3390/molecules28031320_

Round 1

Reviewer 1 Report

1.     It is better to start the abstract with full forms of compounds. Don’t start the abstract with this type of sentence “To modify its affinity for starch”. What do you mean by its affinity?

2.     Clearly define aim of your study.

3.     What is LB agar? Define its role?

4.     How the disproportionation was terminated? What was the standard glucose here?

5.    What is the specification of Langmuir isotherm?

6.    Read the manuscript thoroughly for grammatical and formatting mistakes. Repetition should also be removed.

7.    What was the reason of “all three SBD-TuαGT fusions 348 were relatively less active than TuαGT at > 70 °C, but more active than TuαGT at < 60 349 °C”

8.     Structural analysis of starch before and after modification by TuαGT and SBD-fusions indicated that the fusions with SBDSt1 and SBDSt2 enhanced hydrolysis. Is it beneficial?

9.     What is the future perspective of this study?

10.  There are several grammatical mistakes and syntax errors. They should be removed when revising the manuscript.

Reviewer 2 Report

quite a strange citation of item 1, 21, 22 - no link to the website needed

In my opinion, the purpose of the work was not written clearly enough. It was written more as the results and not the assumptions of the research

Overwhelming workload. very well planned experiences. the research methodology described in detail deserves special praise. Recently, it is very rare for authors to describe their research procedures in such detail - usually it is two or three lines with reference to literature data.

line 232 - instead of the word "powder" she would use either native corn starch or just starch

line 227 -"The Langmuir isotherm .." why this model? single-layer adsorption assuming no intermolecular interactions and adsorption in the form of a monolayer. I know that one can discuss and choose from numerous models, my comment is not in the nature of a questioning assumption, but I would expect an explanation in the methodology.

if the authors on line 75"...was a kind gift of Cargill,..." - on lines 504-505 (Acknowledgments:) the company should be mentioned
